# Motor-Related Mu/Beta Rhythm in Older Adults: A Comprehensive Review

**DOI:** 10.3390/brainsci13050751

**Published:** 2023-04-30

**Authors:** Takashi Inamoto, Masaya Ueda, Keita Ueno, China Shiroma, Rin Morita, Yasuo Naito, Ryouhei Ishii

**Affiliations:** 1Graduate School of Comprehensive Rehabilitation, Osaka Prefecture University, Osaka 583-8555, Japan; precious.honored.i43.4@gmail.com; 2Faculty of Health Sciences, Kansai University of Health Sciences, Osaka 590-0482, Japan; 3Graduate School of Rehabilitation Science, Osaka Metropolitan University, Osaka 583-8555, Japan; ot07092505@gmail.com (M.U.); sambasamba123record@gmail.com (K.U.); sf22199d@st.omu.ac.jp (C.S.); sn23341b@st.omu.ac.jp (R.M.); naitoh@omu.ac.jp (Y.N.); 4Department of Psychiatry, Osaka University Graduate School of Medicine, Suita 565-0871, Japan

**Keywords:** mu rhythm, beta rhythm, aging, older adults, movement, action observation, EEG

## Abstract

Mu rhythm, also known as the mu wave, occurs on sensorimotor cortex activity at rest, and the frequency range is defined as 8–13Hz, the same frequency as the alpha band. Mu rhythm is a cortical oscillation that can be recorded from the scalp over the primary sensorimotor cortex by electroencephalogram (EEG) and magnetoencephalography (MEG). The subjects of previous mu/beta rhythm studies ranged widely from infants to young and older adults. Furthermore, these subjects were not only healthy people but also patients with various neurological and psychiatric diseases. However, very few studies have referred to the effect of mu/beta rhythm with aging, and there was no literature review about this theme. It is important to review the details of the characteristics of mu/beta rhythm activity in older adults compared with young adults, focusing on age-related mu rhythm changes. By comprehensive review, we found that, compared with young adults, older adults showed mu/beta activity change in four characteristics during voluntary movement, increased event-related desynchronization (ERD), earlier beginning and later end, symmetric pattern of ERD and increased recruitment of cortical areas, and substantially reduced beta event-related desynchronization (ERS). It was also found that mu/beta rhythm patterns of action observation were changing with aging. Future work is needed in order to investigate not only the localization but also the network of mu/beta rhythm in older adults.

## 1. Introduction

EEG measures electrophysiological activity in the brain by attaching multiple electrodes on the scalp. In 1929, Berger [1] first reported measurable electric potentials as temporal changes on the scalp. The next year, he found that certain events could disturb the ongoing alpha activity [2]. These phenomena of event-related changes are observed at a specific frequency in the ongoing EEG activity. A dominant alpha activity (8–13 Hz) in posterior regions enhancement occurs when the eyes are closed, and the activity is suppressed by visual stimulation. Alpha rhythm is proposed to represent a cortical “idling” that facilitates cortical activation by different stimuli. The generation of these alpha oscillations is linked to thalamus–cortical interactions [3]. Beta activity (13–25 Hz) changes in the parietal regions during movement-related tasks include movement execution or imagery, and action observation [4]. Increasing delta oscillations (1–4 Hz) in occipital regions appear to occur in patients with prodromal AD, indicating dysfunctional synaptic transmission [5]. An increase in theta power (4–8 Hz) is often interpreted as a slowing of alpha rhythm and correlates with a decline in cognitive processing speed. It is reported that increased theta oscillations in posterior regions may indicate early neurodegeneration [6].

Decreasing power in some frequency bands is known as event-related desynchronization (ERD) and increasing power is known as event-related synchronization (ERS) [7]. EEG and MEG recordings can measure ERD and ERS. MEG measures the weak magnetic fields produced by electrical currents in bundles of synchronized neurons in the cortex [8]. MEG has better spatial resolution than EEG because Magnetic fields are less distorted by the cerebrospinal fluid, sulci, scalp, and skull [9,10,11]. Whereas the advantages of EEG are that it is simple to measure and has significantly lower costs than other methods of studying brain function, both EEG and MEG can measure the responses of cortical activity during resting state and sensory or motor tasks. To understand the mechanisms underlying age-related changes or pathological aging in older adults, neurophysiological methods such as EEG and MEG are useful [12]. EEG and MEG can measure the changes in brain activity during motor tasks immediately (milliseconds), but functional magnetic resonance imaging (fMRI) or positron emission tomography (PET) ordinarily take over 6–7 s to process brain activation because assessment relies on cerebral blood flow [12]. Therefore, it is difficult to analyze the temporal perspective of brain activity during motor tasks using these techniques [13]. High temporal resolution techniques are essential for investigating the brain electric activity changes of cognitive processes with aging [14].

From before to after voluntary movement, upper alpha and lower beta frequency activities are observed [15,16,17,18,19]. The upper alpha band activity is primarily 8–13 Hz and is called “mu” activity. The lower beta band activity is primarily 13–25 Hz but differs between studies. mu/beta ERD would reflect activation from movement preparation to termination [20], and mu/beta ERS indicates a cortical idling state or inhibition of the activation of the sensorimotor cortex [21,22]. However, the definitions of the mu rhythm and beta frequency band are not agreed upon among scientists. Hari [23] claims that the mu rhythm has two main components at 10 and 20 Hz. However, some studies have split the high and low activity as mu and beta rhythms. Additionally, most studies consider only either mu or beta oscillatory activity. The analytic variety creates an issue when using ANOVAs, confirming many interrelating corrections without multiple comparison [24,25]. Hobson and Bishop [25,26] point out the potential functional distinctions between both frequency bands.

Mu/beta activity can be measured from the central areas of the primary sensorimotor cortex by EEG and MEG. For topography between mu and beta rhythms, both frequency components would be generated in the same hand primary motor area, but the mu rhythm is slightly more posterior than the beta rhythm [27]. The neuron source of mu rhythm is considered in the postcentral gyrus related to somatosensory processes. The neuron source of beta rhythm is recognized to be in the precentral gyrus associated with motor functions [28]. This suggests that mu and beta rhythms have distinct functional networks [7,23,27,29,30,31,32]. Beta ERD is related to the synchronized activity in a thalamocortical loop [7].

Before the initiation of voluntary movements or movement observations, mu and beta power attenuation (i.e., desynchronization of mu and beta activity) has begun [33,34]. The ERD initiation begins about two seconds before movement oncoming over the contralateral Rolandic region, then becomes bilaterally symmetrical as it achieves closer execution of movement [7]. This prior movement ERD phenomenon has a relationship with the preparatory activity in the motor cortex [16]. Figure 1 shows a typical example of beta activity flow during voluntary movement (Adopted from Vinding et al. 2019 article [35]).

With the onset of movement, mu ERD occurs bilaterally and widely [36,37], whereas beta ERD becomes bilateral and more restricted [33]. It has been suggested that the desynchronization of both rhythms during voluntary movement or the observation of movements correlates with the activity in the sensorimotor cortex [38]. After movement termination, both activities show ERS for the duration of a few seconds [39]. The beta ERS is observed more clearly than the mu ERS, because the beta ERS shows a power rebound above baseline when the mu rhythm still had a desynchronized pattern of low power. The period of beta ERS post-movement is termed “beta rebound.” Mu ERS in post-movement is interpreted as an active inhibition or deactivation of the neural process [40], whereas beta ERS would relate to “idling” in motor area neurons [39]. The 20 Hz rhythm synchronization after movement corresponds to the period of excitability decreasing corticospinal tracts. Additionally, it supports the idea that ERS is correlated to the idling of neurons in the motor area [41]. The beta rebound is considered to engage in the information process for evaluating the predicted and performed movement [42]. Mu/beta ERD indicates the activation of cortical areas during the performance of the task, and the processing of sensory and cognitive information or production of motor command [3]. Furthermore, increasing ERD is interpreted as the involvement of a larger neural network or more cell assemblies in information processing [7]. Therefore, mu ERD magnitude and size can indicate the neural network size that is recruited for the performance of the task.

Similar activity patterns are also observed during motor images, passive movement, and in observing others’ actions or movements. Mu desynchronization would be associated with the human mirror neuron system (MNS) activity based on results from previous studies [43,44]. For this reason, MNS studies using mu rhythm have increased recently. Moreover, research has been performed on mu study beyond the observation action. According to the summary by Hobson and Bishop [26], it includes speech and language tasks, and social processes such as empathy, theory of mind or biological motion.

Previous studies have mainly evaluated mu/beta rhythms in the context of motor-related tasks. In recent years, mu/beta studies can be categorized into four types. The fundamental research of mu/beta study [45,46], the fundamental research of changes with development in childhood [47,48,49], fundamental research on the effect of training [50,51,52,53,54,55], and clinical research on the effect of training [56,57,58,59]. The subjects of mu/beta rhythm studies range from infants to older adults. Furthermore, these subjects include not only healthy subjects but also patients with various neurological and psychiatric disorders such as Parkinson’s disease [60,61,62] and cerebrovascular diseases [63,64,65], because the mu rhythm is expected to be useful for developing a better intervention for motor rehabilitation in people with disabilities. Mu/beta rhythm modulates the integration of the new ability during motor skill acquisition by practice and is associated with expertise development in a motor-related task [66,67,68]. The subjects in the mu rhythm study using an action observation task are not only healthy people but also those with autism spectrum disorder (ASD) or with Parkinson’s disease [4,62]. Some autism spectrum disorder studies that measured mu rhythm support the idea about the relationship between mu rhythm and MNS. However, most studies have focused on young adults or those with the disease, and very few studies have investigated how the mu rhythm changes with aging. In particular, we have not found any research on mu rhythm in older adults in the last few years.

Aging changes brain structure morphologically and functionally [69,70,71,72,73]. Motor disability was reported to be associated with aging in previous research [74,75,76,77]. The causes of motor disability with aging are multifactorial with cerebral atrophy of the callosum and sensorimotor cortex, the decline of muscles and sensory receptors and peripheral nerves, and the degeneration of neurotransmitter systems [78]. In the non-clinical study of brain activity during motor tasks using fMRI it was observed that older adults showed a wider brain activation than younger people, particularly the prefrontal lobe and basal ganglia circuit loops [78]. Other non-clinical studies also reported that healthy older adults show extended brain activity compared with young people during a simple repetitive hand movement task [79,80]. Furthermore, Mattay et al. [74] investigated the correlation between reaction times and brain activity of motor regions in healthy older adults during a visually paced “button-press” motor task and found an interrelation between shorter reaction times and increasing motor area activity. It has been concluded that the motor area activation represents compensatory mechanisms invoked by the aging brain. In non-clinical studies, Hong et al. [81] also reported that healthy older adults activate wider brain areas to compensate for the decline in cognitive function.

In previous research, the characteristics of brain oscillation in healthy older adults are revealed as electrophysiological changes such as frequency band, power, distribution, and morphology of brain waves. Ishii et al. [12] have summarized that the EEG changes with aging, focus on oscillations, functional connectivity, and signal complexity. Here, we report only EEG characteristics of non-clinical studies in healthy older adults during resting state, which is defined as the deceleration of background activity, a reduction in alpha band amplitude, an increasing delta and theta power [82,83], and an enhanced complexity of dynamic phase synchronization in the alpha band (predominantly frontal brain regions) [84]. In non-clinical studies, Babiloni et al. [85] investigated whether cortical EEG rhythms change with physiological aging using data from over 100 healthy older people. The results showed a decrease in alpha rhythm magnitude with aging, correlated to global cognitive level. Furthermore, Babiloni et al. [86] found alpha components in posterior brain regions decrease with aging. This finding indicates that the power of alpha activity relates to cognitive decline with physiological aging [87]. Vlahou et al. [88] investigated the association between changes in normal aging in brain oscillation and cognitive ability, and reported that higher delta and theta power in the temporal and central regions in older adults has a positive correlation with perceptual speed and executive functioning. Similarly, Finnigan and Robertson [89] suggested that resting theta power in healthy older adults might be correlated with healthy neurocognitive function, not with a decrease in cognitive function. Although beta components are reported to relate to motor-related activity, few studies have reported that resting beta oscillations change with aging.

As seen above, it is known that changes in the brain structure occur with aging, including changes in brain activity and characteristics of brain oscillations, in healthy older adults. As for mu/beta rhythm studies during a move-related task, fundamental research using healthy young adults and clinical research using those with the disease is also performed. However, few studies have been carried out on the mu/beta rhythm in healthy older adults regarding changes with aging. Mu rhythm changes in older adults may be affected by the decline of a sensory nerve and forgotten motor images and experiences.

We are interested in the following two things:(1)How much mu/beta research in older adults using voluntary movement or action observation tasks has been carried out?(2)How does the mu/beta rhythm in older adults show a difference compared to young ones?

The present review aims to outline mu/beta rhythms characteristics in older adults during voluntary movement and action observation compared to young ones. It is important to clarify the findings regarding mu/beta rhythms and their changes with aging until now and promote the study of mu/beta rhythm in older adults for application in rehabilitation for various diseases.

## 2. Method

The article search was conducted using five electronic databases that are widely used in the fields of brain science. These were PubMed, Web of Science, Science Direct, Springer Link, and Google Scholar. The search date was 7–8 May 2022.

### 2.1. Inclusion Criteria

The study used voluntary movement or action observation tasks.The article focuses on mu waves, when measured, and data analysis.The research participants include healthy older groups (aged ≥ 65 years).

### 2.2. Exclusion Criteria

The target population was clinical subjects not including healthy older adults.The study used motor imaging or sensory stimulus task.

### 2.3. Search Terms

The search terms included: (mu rhythm* OR µ rhythm* OR mu waves* OR µ waves*) AND (aging* OR elderly* OR elder* OR older* OR older adults*).

Firstly, one investigator (T.I.) examined the titles and abstract and included 183 articles. Secondly, three investigators (T.I. R.I. and Y.N.) assessed full-text articles to ascertain whether they met the inclusion criteria, and 8 studies were included in the final analysis. The mu/beta rhythm study using the action observation task was 1 out of 8. We tried to summarize them as a scoping review because the number of extracted articles was too small.

## 3. Result

### 3.1. Mu/beta Rhythm during Voluntary Movement in Older Adults

Table 1 shows the review of seven articles about mu/beta rhythm during voluntary movement in older adults. These studies included the subjects from healthy young and older groups.

The extracted seven articles showed roughly four characteristics for mu/beta rhythm in older subjects during voluntary movement.

### 3.2. Mu/beta ERD Amplitude Increase

Older adults have reported that they increased mu/beta ERD. Derambure et al. [17] measured and quantified alpha ERD during a self-paced movement of pressing a microswitch using the thumb and compared to the two age groups. As a result, the older group showed a larger amplitude of ERD compared with the young group before terminating movement. Labyt et al. [91] used three motor tasks (distal, proximal, visuo-guided targeting) and reported that older adults showed increased mu ERD over the ipsilateral regions during motor preparation than the younger ones. Rossiter et al. [94] investigated beta activity changes during the gripping task. It has been reported that aging was related to increased beta ERD in the ipsilateral. Schmiedt-Fehr et al. [95] studied alpha and beta frequency in younger and older subjects during a go/nogo task. The results showed stronger beta ERD during movement preparation and execution in older adults. Mu/beta ERD has been assumed as an indicator of the degree of neuron activity during motor planning [16]. The planning and initiation of movement involves the primary motor areas, the sensorimotor area, basal ganglia, thalamic nuclei, and the cerebellum. Currently, the basal ganglia–cortical system is thought of as reflecting an endogenous top-down control neuron mechanism that internally processes for future events [96,97]. Increasing mu/beta ERD in older adults may reflects over-activation in these areas because increasing ERD indicates that a larger group of cells is involved.

### 3.3. Mu/beta ERD Latency Decrease

Older adults showed that mu/beta ERD began earlier and ended later. A lengthening of alpha ERD duration during movement in older adults was reported [17]. Labyt et al. [92] investigated cortical oscillatory activity during muscular pure contraction and relaxation. Mu/beta ERD’s overlying of the frontocentral and parietocentral in the contralateral area was earlier in older than young ones. Studies by Derambure et al. [17] and Labyt et al. [92] reported no difference in behavior data between the older and younger groups. Vallesi et al. [93] analyzed EEG activity in older adults during a go/nogo task and showed a lengthening of beta ERDs compared with younger ones. The lengthening of mu ERD in this study was associated with longer reaction times with aging.

It should be noted that all motor-related studies in older adults did not exhibit the same results. These inconsistent results seem to have been affected by the multiple factors involved in the type of task and method of analysis for the frequency band. To obtain a more complex picture of those topics, further study of the focus on changes in ERD time window patterns in older adults should be conducted.

### 3.4. Mu/Beta ERD Expansion

Older adults indicate an increase in the recruitment of cortical neurons and become mu/beta ERD more bilaterally. Rossiter et al. [94] reported that older subjects increase beta ERD in the ipsilateral primary motor area during the grip tasks. Vallesi et al. [93] also found mu/beta desynchronization spread more bilaterally in older adults. Mu/beta ERDs becoming more bilateral is considered one of the characteristics of brain oscillation in healthy older adults. Typically, the activation of the ipsilateral primary motor area is suppressed by the contralateral cortex [98,99]. In previous studies of this inhibition mechanism, evidence that motor-related beta bands are associated with GABAergic activity has been reported [100,101,102,103]. GABAergic inhibition increases with aging and affects the sensorimotor area on both sides, especially during movement [94]. Therefore, aging progressively attenuates inhibition by the contralateral primary motor cortex and the ipsilateral deactivation less pronounced. [80,104,105,106]. Derambure et al. [17] found that older subjects showed the spatial spread patterns of alpha ERD overlying the parietal-frontal area during voluntary movement. Labyt et al. [91] reported that more older subjects showed spatial diffusion of mu ERD overlying the parietocentral and parietal regions than young ones during motor planning. Moreover, Labyt et al. [92] reported that older subjects develop mu ERD more widely on both sides preceding muscle contraction and relaxation.

The spatial spread and earlier bilateral in mu/beta ERD in older adults indicate an increase in neuronal activation of sensorimotor areas, indicating less specificity of brain activity regions than in young adults. These changes in the cerebral area with aging reflect the decrease in the efficiency of motor preparation. Wu and Hallett [107] studied spatial changes in ERD with aging by fMRI and have reported that decreasing specific subcortical inputs causes a compensation through the recruiting of larger groups of neurons involved in the cerebral cortex, basal ganglion, and thalamus.

### 3.5. Beta ERS Decrease

Older adults show beta rebound after movement ends that is substantially reduced. Labyt et al. [90] reported beta ERS after movement changes in older adults between the distal, proximal, and targeting movement tasks. In a visuomotor task study by Labyt et al. [91] older adults showed beta ERS more slowly and at only half the power of young ones. Labyt et al. [92] reported beta ERS was significantly lower in older adults than in young subjects after contraction or relaxation tasks. In the go/nogo study by Schmiedt-Fehr et al. [95], young adults showed beta rebound following go-cue, but this was not the case in older adults.

Post-movement beta rebound reflects sensory information processing from afferent somatosensory neurons [108,109]. Labyt et al. [92] reported that, assuming beta rebound relates to somatosensory inputs, the decline of beta ERS in older adults was caused by reafferent sensory inputs reduction and impairment. In other words, the reduction in beta rebound reflects impaired sensory integration. Currently, the beta rebound is considered to be more detailed and related to the processing in the prediction and performance of the movement. Increasing beta ERS promotes the existing state of affairs [110], which means there is no need to change the command set immediately before. Therefore, the reduction in beta rebound in older adults also suggests a prolonged movement evaluation.

### 3.6. Mu Rhythm during Action Observation in Older Adults

The sensorimotor cortex is susceptible to aging and brain activity is extended during movement [79,80,111]. Brunsdon et al. [50] mentioned the paucity of mu rhythm studies carried out during the observed action of adolescents to older adults. He investigated 301 subjects (from age 10 to 86) to find whether mu rhythm changes with aging in the action observation task and found two results. Firstly, alpha band activity grows from adulthood to senility. Secondly, the beta frequency activity declines from over 60 years of age. Dissimilar patterns of alpha and beta activities indicate a distinction in the processing of development with aging. These modulations of alpha and beta activity are associated with a past report of excessive activity in sensorimotor regions in older adults during movement [94,95,112,113]. In other words, he set up a hypothesis from these results that brain over-activation in older adults is compensatory. Note that this is the first study that has measured mu/beta rhythm changes throughout lifespan. One more hypothesis is that older adults show an increase in mu ERD when the observed action reflects the effects of expertise in MNS [67,114,115]. In other words, older adults’ increased activation in the motor area may reflect each adult’s experience and expertise in relation to the observed movement [50]. Cannon et al. [67] investigated whether mu rhythm is affected by movement experience and observation experience, and found that motor experiences showed the largest mu ERD in comparison to observers or novices. These findings lead us to the fact that the sensorimotor mu rhythm is modulated by action expertise that older adults possess much more than younger ones. Presenting similar results, some studies reported that motor experience (dance action) modulates mu rhythm [66,116,117]. Similar results were obtained in fMRI studies. Calvo-Merino et al. [118] reported that expert dancers exhibit the largest activity in premotor and parietal areas when observing their own dance video. According to Brunsdon et al. [50], the experience of various actions in older adults might be increasing with aging because of long daily living.

As described, only a few studies have been conducted on mu rhythm during action observation in older adults. We found only one mu rhythm study using action observation from adolescence to older adults. Currently, few studies have been conducted on the mu rhythm of action observation in older adults.

## 4. Discussion

### 4.1. Brain Over-Activation in Older Adults

We claim that these overall results relate to compensation and a general interpretation of brain over-activation in older adults. The compensation means that additional activation compensates for the decline in cognitive function with aging. The phenomenon of wide cortical activation observed in older adults is consistent with that reported in fMRI studies [78,79,80]. Older adults showed enhanced general activity in the anterior/prefrontal brain region, especially in the pre-frontal and pre-motor areas [74,106,119,120,121]. It was reported that functional reorganization of the central nervous system in older adults brings over-activation in the brain [122]. For example, older adults with slower reaction times did not differ in activation patterns from young adults, but older adults who performed faster showed additional recruitment in motor areas than young ones [74]. Moreover, Wu and Hallet [107] reported that the older subjects indicated a higher increase in the extent of activity in the cerebellar posterior lobe and pre-supplementary motor area when performing automatically some complex motor tasks. The over-activation in the prefrontal and sensorimotor cortical area during motor tasks reflects the additional investment of neuron resources for cognitive control and sensory processing [119,120]. Why does the excessive activation occurs in older adults during motor and cognitive tasks? One hypothesis is the concept of “functional compensation” [123], in which the additional recruitment reflects a compensation for age-related brain degeneration to keep motor performance. Another hypothesis is termed “dedifferentiation”, which proposes that excessive activation may reflect an inability to select appropriate brain regions. Heuninckx et al. [124] tested the two hypotheses to investigate the association of performance on a complex interlimb coordination task and brain activity. The results showed that the older adults observed a correlation between brain activation level and motor performance. This report provided strong evidence for the hypothesis of functional compensation.

Recently, the study of behavioral mechanisms in older people has been addressed, and several studies have supported the compensation hypothesis (for a review see. [125]). Wolpe et al. [126] investigated brain activity measured by fMRI during a force-matching task. The results showed that sensory attenuation was enhanced with aging, and it related to the decrease of grey matter volume and functional connectivity in the pre-sensorimotor area. Hoellinger et al. [127] tested the motor strategies change with aging using moving tasks with an object with a changing weight. The results showed that older adults move objects faster and spend more time accelerating than younger ones. The authors concluded that older subjects rely on predictive processes to compensate for a reduction in sensory sensitivity.

Interestingly, it is reported that the older subjects displayed more activation not only classical motor control regions (primary sensorimotor (SM1), supplementary motor area (SMA) cingulate motor area (CMA), cerebellum) but also additionally recruited more remote regions [124]. The additional activation area is involved with the contralateral superior parietal cortex, contralateral posterior cerebellum, and ipsilateral anterior cerebellum, which is associated with a high-level sensorimotor coordinate and the left frontal lobe (anterior insular cortex, and the inferior frontal gyrus (IFGpo), inferior frontal gyrus, pars opercularis (OpIFG), and superior temporal gyrus), which is related to higher-order auditory processing. It is also involved with the interfacing of internal representations of body parts with external information about motion, visualization strategies to control their movements, and the contralateral frontal lobe (i.e., the pre-dorsal premotor cortex (pre-PMd) and the dorsolateral prefrontal cortex (DLPFC), which relates to cognitive monitoring of movement. These findings are almost consistent with the result of a cognitive study on aging in which activation levels in frontal regions correlated with overall performance in older adults. This phenomenon is often referred to as the “posterior-anterior shift in aging” (PASA) [128] and explains the compensation for dysfunctional sensory-driven bottom-up processing in posterior brain regions with aging. More specifically, it is assumed that older adults may adopt a strategy of increased reliance on predictive processes rather than internal feedback processes because sensory signals become noisy with aging. These findings provide an insight into how we provide motor tasks in older adults, concerning the kind of feedback and instructions given.

However, some older adults exhibit over-activation but not all of the older adults in the above studies. Moreover, some studies have reported that older subjects show brain over-activation unrelated to behavioral data [94,95]. The contradiction of these results can be explained as the alpha and beta rhythms reflect not only motor processes but are also affected by sensory and cognitive processes. Recently, some studies using complex sensorimotor tasks indicate beta oscillations reflect heightened sensorimotor transmission beyond somatosensory. The beta rhythm is affected by visual and memory factors such as visual cue anticipation and processing and short-term memory.

### 4.2. Current Situation of Mu/Beta Rhythm Study in Older Adults

Very few studies have referred to the effect of mu/beta rhythm with aging, and there was no literature review about this theme. We could find only eight mu/beta rhythm studies that used voluntary movement or action observation in older adults. Mu rhythm studies using infant subjects focused on revealing the brain activation or MNS from the perspective of developmental aspects and many findings are reported. Moreover, many mu rhythm studies are carried out for understanding the motor impairment motor mechanism or MNS and indexing its effect on the rehabilitation of various diseases, such as autism spectrum disorder [129,130], Parkinson’s disease [35], complete spinal cord injury [131], and cerebrovascular disorders [132]. However, not many fundamental studies have been carried out on mu rhythm in older adults. We claim that almost all mu rhythm studies have focused on research applied as above and have forgotten to investigate the basic research that mu rhythm changes with aging, and that understanding the developmental alteration of the motor mechanism and MNS is meaningful.

In the future, it is important to further examine the stages of cerebral activation during movement or action observation in older adults. However, the temporal resolution of fMRI is limited and inadequate for investigating the neuronal activity of a motor task in different phases (preparation, execution, post-movement). EEG and MEG have good temporal resolution and it will be useful to elucidate these issues. Additionally, it is necessary to investigate the interaction between oscillatory modes and the effects of other frequencies for mu rhythm.

Additionally, we can speculate that it is necessary to investigate not only the localization but also the network of brain activity. Functional connectivity is an index of the integration between neural populations, and it can confirm the temporal synchrony and correlation between signals of two or more spatially separated brain regions. Regarding Alzheimer’s disease, we reported on several studies on the functional connectivity of resting-state EEG [133,134]. These methods of nonlinear functional connectivity will be useful for a better understanding of motor-related and action observation mu/beta activity in older people.

## 5. Conclusions

This review focused on changes in mu/beta rhythm with aging during motor-related tasks and action observation. Mu/beta rhythm studies in older adults, especially action observation studies, were much fewer in number than studies in young adults and infants. By reviewing these studies, we found that older adults showed mu/beta activity changes that fall under four characteristics during voluntary movement compared with young adults, such as ERD increasing, an earlier beginning and later end, a symmetric pattern of ERD and increased recruitment of cortical areas, and a substantially reduced beta rebound. We suggest that it represents the compensation that enhanced the investment of neural resources to sustain better motor performance. It was also found that mu/beta rhythm patterns of action observation are changing with aging, and that mu/beta rhythms have distinct developmental trajectories. Additionally, we found several studies reporting that older adults showed increased recruitment of cortical areas. Older adults are the highest proportion to be subject to rehabilitation. In older adults, revealing the motor-related mu/beta rhythm related to action learning, imitation, and long-term effects of action observation will offer fundamental knowledge for better task presentation of rehabilitation intervention.

## Figures and Tables

**Figure 1 brainsci-13-00751-f001:**
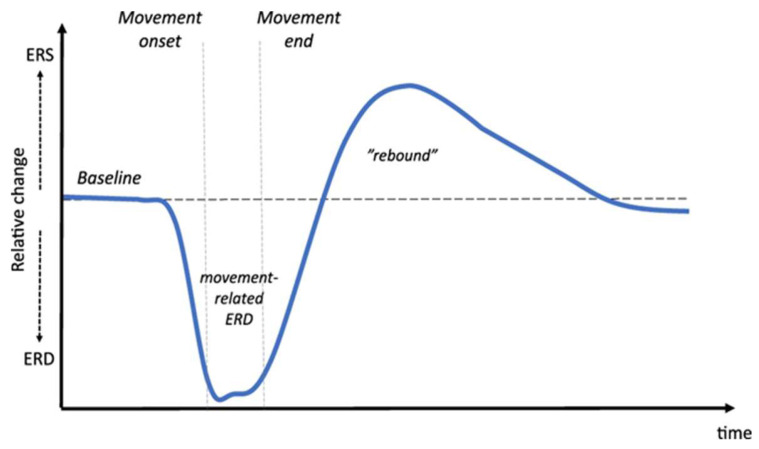
A typical example of Mu-beta ERD/S flow during voluntary movement. (Adopted from Vinding et al. 2019 article [35]).

**Table 1 brainsci-13-00751-t001:** The summary of seven articles about mu/beta rhythm during voluntary movement in older adults.

Authors	Task	Frequency	Result (The Characteristics in Older)	Discussion
P. Derambure et al. (1993) [17]	Press a micro switch with the thumb.	Alpha: 9–13 Hz	Larger ERD amplitude. Earlier began and ended later ERD. ERD spatial diffusion ERD bilateralization.	1. Elderly subjects could correspond to a less specific cortical activation. 2. The desynchronization recorded over the fronto-central region seems to reflect the need for neuronal activation of sensorimotor area. 3. Global increase in the ERD could reflect the need for neuronal activation of Sensorimotor area and changes of input from sensory-motor cortex—thalamus and basal ganglia to sensorimotor area.
E. Labyt et al. (2003) [90]	Distal and proximal and targeting movement.	Mu: 8–12 ± 1 Hz Beta: 13–25 Hz	Decrease of beta ERS. Diffusion of beta ERS.	The beta ERS changes observed can be explained by the sensory processing of various afferent inputs and suggests that sensory integration decreases with age.
E. Labyt et al. (2004) [91]	Distal and prox-imal and targeting movement.	Mu: 8–12 ± 1 Hz Beta: 13–25 Hz	Increases ERD over ipsilateral regions. The spatial extent of ERD over parietocentral and parietal regions. Decrease of beta ERS after the movement.	1. The temporo-spatial changes in ERD patterns with age might originate from a loss of specificity of subcortical inputs and could reflect less efficient motor planning. 2. The beta ERS changes with aging may reflect a decrease in reafferent sensory inputs and impaired sensory processing.
E. Labyt et al. (2006) [92]	Muscular relaxation and contraction.	Mu: 8–12 ± 1 Hz Beta: 13–25 Hz	More widespread and bilateralization mu/beta ERD preceding muscular relaxation. Mu/beta ERD began earlier. Attenuated the beta ERS.	1. The spatial diffusion and earlier bilateralization of ERD might reflect compensatory mechanisms which lead to the correct motor program. 2. The reduction in beta ERS might reflect a decrease in reafferent sensory inputs and their impaired processing. 3. This symmetric pattern of motor cortex activation can be due to a compensatory mechanism for the progressive loss of cortico-spinal motoneurons with age, or for a decrease of long-range connectivity with other regions of the motor loops.
A. Vallesi et al. (2010) [93]	Three condition go/nogo task. (go, high-conflict nogo, low-conflict nogo)	Mu: 10 Hz Beta: 18–22 Hz	The symmetric pattern of mu/beta ERD. Ended later beta ERD.	The pre-activation both motor regions and a longer period of time might be due to a loss of inter-hemispheric inhibition.
E. Rossiter et al. (2014) [94]	Simple unimanual grip task.	Beta: 15–30 Hz	Greater beta power during rest task. Increases beta ERD and reduced beta frequency during movement.	These findings suggest greater GABAergic inhibitory activity within motor cortices of elderly adults.
Schmiedt-Fehr et al. (2016) [95]	The auditory go/ nogo task.	Alpha1: 8–10 Hz Alpha2:10–13 Hz Beta1: 14–21 Hz Beta2: 22–30 Hz	Earlier and more prominent beta-2 decrease before and during movement. Absence of beta-1 and beta-2 rebound. Additional rebound in alpha power after nogo task	1. Increases beta ERD would indicate that they traded accuracy for speed. 2. Reduced or absent beta ERS could be related to prolonged movement evaluation, anticipation of the upcoming feedback tone, and an enhanced disposition for adaptation. 3. The stronger attenuation of beta activity and the reduced beta-rebound most likely indicates neural over-recruitment in healthy aging, which is related to with alterations in multiple factors associated with sensory and cognitive aspects of motor control.

## Data Availability

All data generated or analyzed during this study are included in this article. Further enquiries can be directed to the corresponding author.

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
