# Peer review of "Motor-Related Mu/Beta Rhythm in Older Adults: A Comprehensive Review"

_brainsci, 2023, doi:10.3390/brainsci13050751_

Round 1

Reviewer 1 Report

Dear Editor,

Thank you for asking me to take on the role of reviewer.

The review examined and described the details of the characteristics of mu/beta rhythm activity in older adults compared with young adults, focusing on age-related changes in mu rhythm. The authors found that compared with young adults, elderly adults exhibited changes in mu/beta activity in four features during voluntary movements, including increased event-related desynchronization (ERD), earlier onset and later termination, a symmetrical pattern of ERD and increased recruitment of cortical areas, and significantly reduced beta rebound. It was also found that the mu/beta rhythm patterns of action observation changed with aging.

Overall, the review covers an interesting topic, but the presentation and discussion of the results could be presented more clearly and efficiently. Furthermore, no information on the method used for the selection of literature is mentioned. I found some (minor) grammatical problems, probably related to English being a second language for the authors.

My main concerns relate to the following issues.

Reviewer's comments

Abstract

·       In the abstract (and in the introduction) it should be pointed out that the sensorimotor EEG activity at rest consists of mu waves (8-13 Hz, in the same frequency range as alpha waves).

Introduction

·       In the introduction, especially in the first part, much more recent literature on the subject should be given.

·       In the introduction, in the part on brain oscillations, it is essential to indicate whether the results of the studies mentioned were related to non-clinical (e.g., Hon et al., 2016) or clinical subjects (e.g., Ishii et al., 2017).

Other sections

·       I think the organization of the manuscript is somewhat confusing. What the authors call chapters, and what I would recommend be labeled paragraphs, I think should be part of the introduction. The authors report in “chapters” 2, 3, 4 a description of EEG and MEG techniques and the difference between mu and beta found in previous studies in each type of population (studies in monkeys are also referenced). This is not consistent with the aim of the review, which should be mentioned just before the current chapter 5. The comprehensive review summarizes information on a topic to provide an overview of the available literature but on a particular subject. In my opinion, the topic must be narrowed in terms of the goals the review is intended to achieve.

·       I would like to know from the authors how they searched for articles, what methodology was used to collect and select information and articles. Did the articles selected to study mu/beta rhythm in older adults refer only and exclusively to a healthy population? From year the authors did they begin searching for articles/works to examine the literature? For example, the following work was not included is: Kanokwan S, Pramkamol W, Wipatcharee K, Warissara W, Siwarit R, Sompiya S, Onuma B, Mitra S. Age-related differences in brain activity during physical and imagined sit-to-stand in healthy young and older adults. J Phys Ther Sci. 2019 May;31(5):440-448. doi:10.1589/jpts.31.440.

·        Finally, the results and conclusions do not capture the differences found when comparing older and young adults (differences mentioned in the abstract i.e., increased event-related desynchronization (ERD), earlier onset and later termination, symmetrical pattern of ERD and increased recruitment of cortical areas, and significantly reduced beta rebound). In the conclusions and in chapter 5, authors described what happens in the elderly people with respect to the mu/beta rhythm. It would be useful to clarify the differences (purpose of the review), perhaps by a table comparing the two groups and the main findings associated with them.

Author Response

Thank you for your letters and for the reviewers’ comments concerning our manuscript entitled “Motor-related mu/beta rhythm in older adults: a comprehensive review.” (Manuscript ID; brainsci-2322410). All these comments were important and helped us revise and improve the paper. We have carefully read the comments and made modifications. Changed parts are written in red.

Responses to Reviewer 1

1.       In the abstract (and in the introduction) it should be pointed out that the sensorimotor EEG activity at rest consists of mu waves (8-13 Hz, in the same frequency range as alpha waves)

We agree with reviewer about the lack of description regarding mu waves in the abstract. In the first part of the abstract, we described a brief outline of the mu wave and clearly defined the mu frequency band. We added the following text to the first line in the abstract.

“The mu rhythm, as known as the mu wave, is occurring on sensorimotor cortex activity at rest, and the frequency range is defined 8-13Hz, same frequency as the alpha band.”

2.       In the introduction, especially in the first part, much more recent literature on the subject should be given.

Thank you for pointing out these important points. We added the latest mu/beta studies in recent literature after the measurement techniques and mu/beta fundamental knowledge paragraphs in the introduction.

(Page 3, Line 127; The following was the manuscript)

“Previous studies have mainly evaluated mu/beta rhythms in the context of motor-related tasks. In recent years, mu/beta studies can be categorized into four types. The fundamental research of mu/beta study [45,46], the fundamental research of changing with development in childhood [47–49], and the fundamental research on the effect of training [50–55], the clinical research on the effect of training [56–59]. The subjects of mu/beta rhythm studies range from infants to the older adults. Furthermore, these subjects include not only healthy subjects but also patients with various neurological and psychiatric disorders such as Parkinson's disease [60–62] and cerebrovascular diseases [63–65].”

3.       In the introduction, in the part on brain oscillations, it is essential to indicate whether the results of the studies mentioned were related to non-clinical (e.g., Hon et al., 2016) or clinical subjects (e.g., Ishii et al., 2017). 

We appreciate reviewer for pointing out that we did not provide information whether the subjects are non-clinical or clinical in each cited study and we agree this is an important omission. 

This part is all non-clinical study because it summarizes the results of changes in brain oscillations with normal aging. However, we did not state this explicitly you pointed out, so we added the manuscript of the type of research (non-clinical) or the attributes of the participants (healthy older adults) to make it clearer whether it was brain clinical research. Furthermore, we checked throughout the manuscript to do not get confused due to this pointing out is not limited to the part on brain oscillations.

4.         I think the organization of the manuscript is somewhat confusing. What the authors call chapters, and what I would recommend be labeled paragraphs, I think should be part of the introduction. The authors report in “chapters” 2, 3, 4 a description of EEG and MEG techniques and the difference between mu and beta found in previous studies in each type of population (studies in monkeys are also referenced). This is not consistent with the aim of the review, which should be mentioned just before the current chapter 5. The comprehensive review summarizes information on a topic to provide an overview of the available literature but on a particular subject. In my opinion, the topic must be narrowed in terms of the goals the review is intended to achieve.

We agree with reviewer about the organization of the manuscript. To adhere to a specific format (introduction, method, result, discussion) for narrative review, we moved chapters 2, 3, and 4 to be part of the introduction. Additionally, we tackled deleting descriptions that have less relevance to the theme of this paper (aging-related changes in mu/beta), summarized them into paragraphs more briefly, and explained them in order. In particular, the sentences on topics different from the purpose of this review (studies in monkeys, an argument of action observation mu rhythm) were deleted. We remained the explanation about the fundamental knowledge necessary for mu/beta study in older adults such as the knowledge of the temporal and spatial changes in the mu/beta rhythm.

5.       I would like to know from the authors how they searched for articles, what methodology was used to collect and select information and articles. Did the articles selected to study mu/beta rhythm in older adults refer only and exclusively to a healthy population? From year the authors did they begin searching for articles/works to examine the literature? For example, the following work was not included is: Kanokwan S, Pramkamol W, Wipatcharee K, Warissara W, Siwarit R, Sompiya S, Onuma B, Mitra S. Age-related differences in brain activity during physical and imagined sit-to-stand in healthy young and older adults. J Phys Ther Sci. 2019 May;31(5):440-448. doi:10.1589/jpts.31.440.

We really appreciate reviewer that we did not state this explicitly. We added a methodology section with search methods, inclusion and exclusion criteria, search terms, and the protocol for extracting the article to review. As you pointed out, this review included healthy older populations and the study using movement execution or action observation tasks. We added the following to the manuscript in the methodology section:

(Page5, Line: 203; The following was the manuscript).

“The article search was conducted using five electronic databases that are widely used in the fields of brain science. There were PubMed, Web of Science, Science Direct, Springer Link, and Google Scholar. The search date was May 7-8, 2022. 

Inclusion criteria

The study used voluntary movement or action observation tasks.

The article focuses on mu waves when measured and data analysis.

The research participants include healthy older groups (aged ≥ 65 years).

Exclusion criteria

The target population was clinical subjects or not including healthy older adults.

The study used motor imaging or sensory stimulus task.

Search terms

The search terms included: (mu rhythm* OR µ rhythm* OR mu waves* OR µ waves*) AND (aging* OR elderly* OR elder* OR older* OR older adults*).

First, one investigator (T.I.) examined the titles and abstract and included 183 articles. Second, three investigators (T.I. R.I. and Y.N.) assessed full-text articles whether meet inclusion criteria, and 8 studies were included in the final analysis. The mu/beta rhythm study using the action observation task was 1 out of 8.”

Thank you for your suggestion about the literature. The literature you suggested could not be included in the review because did not meet inclusion criteria due to using motor imagery task.

6.         Finally, the results and conclusions do not capture the differences found when comparing older and young adults (differences mentioned in the abstract i.e., increased event-related desynchronization (ERD), earlier onset and later termination, symmetrical pattern of ERD and increased recruitment of cortical areas, and significantly reduced beta rebound). In the conclusions and in chapter 5, authors described what happens in the elderly people with respect to the mu/beta rhythm. It would be useful to clarify the differences (purpose of the review), perhaps by a table comparing the two groups and the main findings associated with them.

We agree with reviewer and incorporated this suggestion in our paper.  In part of the results, we included a table attached about the differences in mu/beta rhythm in older adults when comparing young adults (Line; 232, see Table 1). Table 1 showed detailed information on previous studies (authors, the task used, interested frequency band, result, and discussion). Additionally, we added the manuscript with the outline of Table 1 and the subject information of reviewed study. The following was added to the manuscript as part of the results.

(Page 5, Line; 228) “Table 1 showed the review of seven articles about mu/beta rhythm during voluntary movement in older adults. These studies included the subjects from healthy young and older groups.”

(Page 7, Line; 236) “The extracted seven articles showed four characteristics roughly for mu/beta rhythm in older subjects during voluntary movement.”

We believe the table (The mu/beta rhythm characteristics specific to the older compared to the young) became clarify the purpose of the review.

Reviewer 2 Report

Overall impression:

The review tackles an interesting topic, the specific bandwidth frequencies of resting MU and Beta rhythms in the context of older adults compared to young. Furthermore, the subject matter may be of relevance in neurorehabilitation like action-observation therapy where parameters such as event-related desynchronization could be used as a marker for adaptations. Additionally, the parts that describes the origin and interpretation of these techniques, parameters and physiological impact are detailed and easy to follow, which we believe the author should be credited for.

However, the article does not express what sort of review this. Perhaps a scoping or narrative review format would be suitable. Overall structure of the article needs quite a bit of work in relation to a specific format, in addition to explicitly stating the aim of the review. We also believe the authors would benefit from addressing the English language throughout the text with the help of a native speaker.

We have outlined some more specific comments below that may help the authors to address these points.

·        In the introduction the various bandwidth frequencies that are discussed throughout the text are not clearly defined. The authors refer to Ishii and co-workers (Line: 47). However, the reader should not have to read another review to understand one of the main points continually addressed throughout the article (alpha, beta, theta power). Please include a brief definition for these frequencies and what their physiological basis is suggested to be.

·        Some abbreviations like EEG are not defined or are defined after they are first mentioned as an abbreviation. For example, in line 83 MEG is mentioned but is not defined until line 103.

·         

·        Line 49: Does the author mean "defined as" when "reported as"? This is important the definition of resting-state EEG characteristics in elderly will be tied to your aim. We believe "defined as" is more appropriate here.

·        Line 58-59: Please try not to use the term ”they" and instead state the author to avoid confusion.

·        Line 71: The English here needs to be addressed. Perhaps re-phrase this as: "Previous studies have mainly evaluated MU/rhythms in the context of xxx"

·        79-82: The author needs to express the aim more explicitly and be more specific. Example: "The present review aims to outline resting state EEG characteristics of MU/beta rhythms compared to young"

·        136-137: The authors cover an interesting hypothesis regarding ERD and how it is related to recruitment of larger cortical areas to prepare for a task.

·        197-198: This should have been addressed in the introduction. This means that any study discussed beyond the introduction should include these two frequencies.

·        241: Instead of numbers below this heading, I believe that these should be subheadings. For example: ERD increase, ERD latency, Cortical activation characteristics.

·        260-263: While this is explained nicely, it would have been interesting to know the author's interpretation instead of simply stating what has been said before, this occurs in a few places throughout the article.

Overall the article tackles an interesting topic and covers the specific techniques and parameters quite well, in addition to discussing the findings in the literature. However, the English language needs work and the review would benefit from adhering to a specific format like a scoping or narrative review (see. SANRA) . We also encourage the author to discuss their findings from their own point of view and how it may relate to the aim.

Author Response

Responses to Reviewer 2

  1. We believe the authors would benefit from addressing the English language throughout the text with the help of a native speaker.

Thank you for pointing out these important points. Regarding the revision of English, the manuscript was improved the grammar and stylistic expression of the paper by an experienced scientific editor. we paraphrased the “elderly” to “older” throughout the manuscript. However, we are willing to make more changes if you think warrants revision.

  1. In the introduction the various bandwidth frequencies that are discussed throughout the text are not clearly defined. The authors refer to Ishii and co-workers (Line: 47). However, the reader should not have to read another review to understand one of the main points continually addressed throughout the article (alpha, beta, theta power). Please include a brief definition for these frequencies and what their physiological basis is suggested to be.

We agree with reviewer about these points. We defined each frequency band (alpha, beta, theta power) and added a brief physiological basis at the beginning of the introduction. Mu and beta bands were described in the abstract and explanation of fundamental knowledge of mu/beta rhythm (the third paragraph in the introduction).

(Page1, Line: 36)

“A dominant alpha activity (8-13Hz) in posterior regions enhancement occurs when the eyes are closed, and the activity is suppressed by visual stimulation. Alpha rhythm is proposed to represent a cortical “idling” that facilitates cortical activation by different stimuli. The generation of these alpha oscillations is linked to thalamus-cortical interactions [3]. Beta activity (13-25Hz) changes in the parietal regions during movement-related task including movement execution or imagery, and action observation [4]. Increasing delta oscillations (1-4Hz) in occipital regions appear to occur in patients with prodromal AD, indicating dysfunctional synaptic transmission [5]. An increase in theta power (4-8Hz) is often interpreted as a slowing of alpha rhythm and correlates with a decline in cognitive processing speed. It is reported that increased theta oscillations in posterior regions may indicate early neurodegeneration [6].”

  1. Some abbreviations like EEG are not defined or are defined after they are first mentioned as an abbreviation. For example, in line 83 MEG is mentioned but is not defined until line 103.

We thank reviewer for pointing that we did not define some abbreviations used in the manuscript. We checked used abbreviations throughout all manuscripts and defined abbreviations where they were first mentioned (it included EEG, MEG, fMRI, PET, ERD, ERS, ASD, and MNS).

  1. Line 49: Does the author mean "defined as" when "reported as"? This is important the definition of resting-state EEG characteristics in elderly will be tied to your aim. We believe "defined as" is more appropriate here.

Thank you for your advice. We changed as pointed out to tie to our aim. (Line: 167)

“Here, we report only EEG characteristics of non-clinical studies in healthy older adults during resting-state, which is defined as the deceleration of background activity, the reduction of alpha band amplitude, and increasing delta and theta power.”

  1. Line 58-59: Please try not to use the term” they" and instead state the author to avoid confusion.

Thank you for your suggestion. We altered the sentence as below.

(Page 4, Line: 177)

“Vlahou et al. [88] investigated the association between changes in normal aging in brain oscillation and cognitive ability, and reported that higher delta and theta power in the temporal and central regions in older adults was a positive correlation with perceptual speed and executive functioning.”

  1. Line 71: The English here needs to be addressed. Perhaps re-phrase this as: "Previous studies have mainly evaluated MU/rhythms in the context of xxx"

Thank you for your specific advice on English expressions. We have rephrased to better convey our intentions as per your advice.

(Page 3, Line: 127)

“Previous studies have mainly evaluated mu/beta rhythms in the context of motor-related tasks.”

  1. 79-82: The author needs to express the aim more explicitly and be more specific. Example: "The present review aims to outline resting state EEG characteristics of MU/beta rhythms compared to young"

We agree with reviewer’s comments. We rephrased the sentence about the aim of this review to more explicitly and be more specific.

(Page 5, Line: 198)

“The present review aims to outline mu/beta rhythms characteristics in older adults during voluntary movement and action observation compared to young ones.”

  1. 197-198: This should have been addressed in the introduction. This means that any study discussed beyond the introduction should include these two frequencies.

Thank you for your suggestion. We made significant changes to the introduction to conform to a specific format. Part of the discussion about the disagreement of the mu (and beta) rhythm definition was included in the overview of the mu/beta section in the introduction (second paragraph, Line; 70).

  1. 241: Instead of numbers below this heading, I believe that these should be subheadings. For example: ERD increase, ERD latency, Cortical activation characteristics.

Per your request, we have changed the from numbers heading to subheadings.

(Page 7, Line: 239)

From (1) to “Mu/beta ERD amplitude increase

(Page 8, Line: 257)

From (2) to “Mu/beta ERD latency decrease”

(Page 8, Line: 273)

From (3) to “Mu/beta ERD expansion”

(Page 8, Line: 299)

From (4) to “Beta ERS decrease

  1. 260-263: While this is explained nicely, it would have been interesting to know the author's interpretation instead of simply stating what has been said before, this occurs in a few places throughout the article.

Thank you for the opportunity to clarify this point. We acknowledge that the organization of our manuscript was somewhat confusing. We have organized the manuscript (adhering to the specific format as introduction, method, result, and discussion) and clarified our interpretation in part of the discussion. Our interpretation is that the four characteristics of mu/beta activity in older adults (extracted from the review) may reflect brain hyperactivation, and it represents the compensation that enhanced the investment of neural resources to sustain better motor performance. It is the world's first mention in the world that to analyze the brain activity of older adults during motor-related tasks and found the phenomenon of brain hyperactivation from the perspective of mu rhythm. Our interpretation of the results was described in the first line of the discussion and the conclusions, but we have small changed the manuscript for more emphasis.

Reviewer 3 Report

Dear Authors,

First of all, I would like to congratulate you for the large number of references used, which give great support to your study.

Manuscript shows the activation of mu / beta rhythm in elderly trough movement, action observation. To know brain activation is an important topic to establish treatments which without a doubt may be present in advanced ages.

Manuscript has synthetized the latest evidence about this topic in a narrative review structured in 6 chapters.  As a suggestion it would be very useful to add a line time about the news in this topic regarding the selected articles you have and also a comparison table with the results of these manuscripts in each chapter.

Although it is a narrative review, a methodology section must be added: which databases you have used, research time, inclusion/exclusion criteria followed, selection of studies….

Conclusions are supported by the data presented.

Author Response

Responses to Reviewer 3

  1. As a suggestion it would be very useful to add a line time about the news in this topic regarding the selected articles you have and also a comparison table with the results of these manuscripts in each chapter.

We agree with reviewer and incorporated this suggestion in our article. We have included a table attached to mu/beta rhythm differences found in older adults compared to young adults. It showed the data that authors, the task used, interested frequency band, and summarized the result and discussion from previous studies (see Table 1).

  1. Although it is a narrative review, a methodology section must be added: which databases you have used, research time, inclusion/exclusion criteria followed, selection of studies….

As you pointed out, we have added the method section and described search methods, inclusion and exclusion criteria, search terms, and the protocol for extracting the article to review.  We added the following to the manuscript in the methodology section:

(Line: 203; The following was the manuscript).

“The article search was conducted using five electronic databases that are widely used in the fields of brain science. There were PubMed, Web of Science, Science Direct, Springer Link, and Google Scholar. The search date was May 7-8, 2022.

Inclusion criteria

The study used voluntary movement or action observation tasks.

The article focuses on mu waves when measured and data analysis.

The research participants include healthy older groups (aged ≥ 65 years).

Exclusion criteria

The target population was clinical subjects or not including healthy older adults.

The study used motor imaging or sensory stimulus task.

Search terms

The search terms included: (mu rhythm* OR µ rhythm* OR mu waves* OR µ waves*) AND (aging* OR elderly* OR elder* OR older* OR older adults*).

First, one investigator (T.I.) examined the titles and abstract and included 183 articles. Second, three investigators (T.I. R.I. and Y.N.) assessed full-text articles whether meet inclusion criteria, and 8 studies were included in the final analysis. The mu/beta rhythm study using the action observation task was 1 out of 8.”

Round 2

Reviewer 1 Report

Although some critical issues remain, the authors have addressed the issues and improved their manuscript overall